# FRACTAL3DGEN: COARSE-TO-FINE 3D GENERATION VIA FRACTAL HIERARCHICAL TRANSFORMERS

## ABSTRACT

Autoregressive generation models have demonstrated strong performance across a wide range of applications. In the context of 3D shape generation, several approaches have attempted to adopt this paradigm by flattening the 3D representation into a serialized one-dimensional sequence and predicting it sequentially, token by token. These methods are inherently inefficient due to their strictly sequential generation process and are often inadequate in exploiting hierarchical self-similar representations during generation. Inspired by the success of FractalGen in image generation, we propose *Fractal3DGen*, a hierarchical fractal-autoregressive framework that leverages self-similarity across multiple scales for 3D shape generation. Specifically, the 3D shapes, represented by sparse SDF grids, are encoded into a compact, low-resolution sparse voxel latent space for more efficiency. Our generation process then employs a divide-and-conquer strategy to efficiently capture the intrinsic hierarchical structure of 3D shapes. Furthermore, due to the spatial sparsity of 3D shapes, we introduce an adaptive pruning mechanism during generation to promptly halt the processing of empty regions at higher scales, significantly improving both training and inference efficiency while reducing memory consumption. Extensive experiments on ShapeNet demonstrate the effectiveness of Fractal3DGen. Fractal3DGen achieves the lowest average FID against all baselines, with over 15% improvement across five categories. Moreover, it provides a 1.85× speedup, reducing the average inference time per shape from 54.1s to 29.3s compared with OctGPT. Beyond benchmark evaluations with class conditioning, we further explore text- and image-conditioned 3D generation, demonstrating the generalizability of our approach.

## 1 INTRODUCTION

The advancement of 3D generative models has become increasingly critical, driven by growing demands for high-quality 3D content in gaming, film production, and industrial design. Although autoregressive models have demonstrated remarkable capabilities in areas such as large language models (LLMs), efficiently and effectively adapting them for 3D shape generation remains a challenging and worthwhile topic due to the complexity and sparsity of 3D shapes.

Existing autoregressive 3D generation methods can be broadly categorized into two types. One type (Siddiqui et al., 2024; Wang et al., 2024; Weng et al., 2024) directly and sequentially predicts mesh triangle faces. Due to the lack of effective representation compression, these methods struggle to handle very long sequences of faces, which limits their ability to generate high-quality 3D shapes. In contrast, other methods (Zhang et al., 2022; Mittal et al., 2022; Wei et al., 2025) compress the 3D shape into a compact latent space and perform the generation process within this latent space, achieving promising results. However, these methods, which rely on a token-by-token prediction paradigm, struggle to capture the intrinsic hierarchical structure of 3D shapes and the self-similarity across multiple scales.

Recent autoregressive-based 3D generation models (Ibing et al., 2023; Zhou et al., 2023; Wei et al., 2025) introduce multi-scale into the latent 3D representation and generation process. OctGPT (Wei et al., 2025) adopts a hierarchical octree-based structure 3D representation and performs predictions by flattening all levels across the octree into a long sequence, generating tokens in a token-by-token

manner. Despite employing multiple token prediction, it still requires a long time to generate good results for long sequences.

Inspired by FractalGen (Li et al., 2025) and the intrinsic hierarchical structure of 3D shapes, we, for the first time, introduce fractal theory (Mandelbrot, 1998; Goldberger et al., 2002) into 3D shape generation. Fractal3DGen adopts a hierarchical fractal-autoregressive architecture, in which the fractal generator serves as the fundamental self-similar generative unit. At each level, the fractal generator progressively refines voxel blocks through a Transformer module, enabling coarse-to-fine prediction and end-to-end generation from low to high resolution (see Fig. 1). This design offers two key advantages:

1. **Hierarchical Generation:** Fractal3DGen first captures the global structure and then progressively refines local details. This coarse-to-fine process shortens the effective sequence length handled by the Transformer.

2. **Divide-and-Conquer:** Fractal3DGen decomposes the complex generation process into split-and-refine operations for each voxel node, allowing each node to independently focus only on the features from its parent level. This reduces the learning difficulty of the generator module and enables the resulting sub-tasks to be executed independently, thereby increasing the parallelism of the generation process.

Another challenge in 3D generation is that most 3D shapes are largely empty. Existing AR models (Ibing et al., 2023; Zhang et al., 2022; Xiong et al., 2024; Zhang et al., 2024; Zhou et al., 2023; Wei et al., 2025) either treat all voxels uniformly or rely on specialized structures such as octrees, with networks tailored to these representations. These approaches results in complex architectures that are hard to scale and deploy. Fractal3DGen addresses this issue with a pruning mechanism (Hastie et al., 2009; Han et al., 2015), illustrated by the transparent voxel blocks in Fig. 1. Operating on structured voxel grids, it prunes directly on points or voxels, simplifying both the network and 3D representation while exploiting the inherent sparsity of 3D objects for efficient generation.

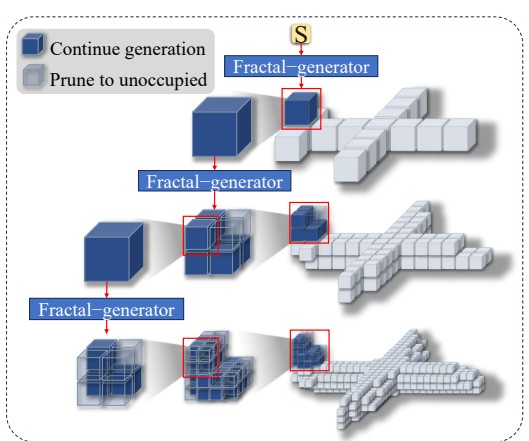

Figure 1: **The Fundamental Fractal Principles** Fractal3DGen recursively generates and refines voxel blocks through self-similar fractal generators, progressively capturing global structure and local details from coarse to fine resolutions.

On the ShapeNet benchmark (Chang et al., 2015), Fractal3DGen achieves a single-sample generation time of 29.296 s, a 1.85× speedup over the OctGPT baseline (54.083 s), while delivering superior quality with an average FID of 27.85, outperforming OctGPT (32.90) and Oct-Fusion (33.24). Its adaptive spatial pruning attains 77.6% efficiency, and the generated models maintain a stable average Hausdorff dimension of 2.272, highlighting the preserved fractal characteristics.

The key contributions of this work are summarized as follows:

1. We propose a novel 3D shape generation model that pioneers the integration of fractal theory into 3D generation, substantially improving both the quality and efficiency of autoregressive models.

2. We present a 3D fractal pruning strategy for efficient inference, and propose a normalized Hausdorff dimension to quantify model complexity and evaluate pruning effectiveness.

3. Fractal3DGen achieves significant improvements in generation speed, quantitative metrics, and computational efficiency, demonstrating both higher inference efficiency and superior generative quality compared to existing methods.

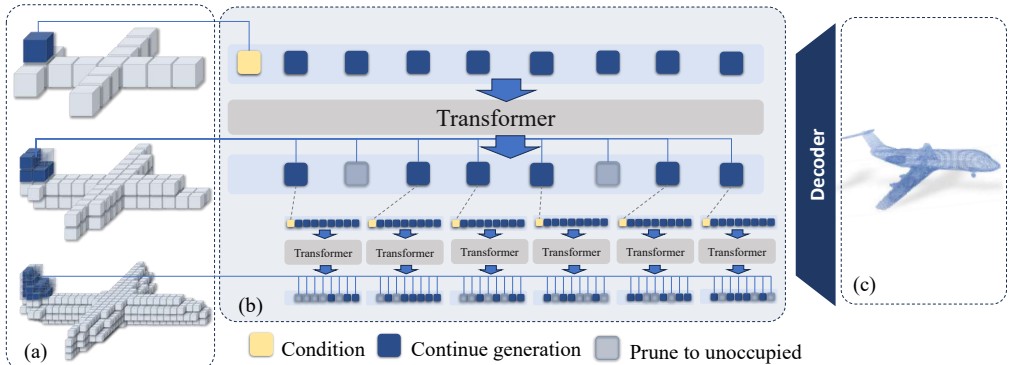

Figure 2: **3D Fractal Generation Architecture.** (a) shows the coarse-to-fine 3D shape generation process. Using the blue part of the aircraft tail as an example, (b) illustrates the corresponding fractal generation structure. At each level, the model predicts the next-level voxel block representations via self-attention, which serve as conditions for subsequent generation. It also predicts voxel block occupancy: Dark blue blocks continue to the next level, while light blue blocks are pruned. (c) shows the final 3D shape at target resolution, obtained by decoding the latent representation with a pre-trained VQ-VAE.

## 2 RELATED WORK

### 2.1 FRACTAL

In recent years, the application of fractal theory in the field of computer vision has gradually gained attention. The fractal geometry proposed by Mandelbrot (1967) provides a theoretical foundation for describing complex self-similar structures in nature. In the field of deep learning, fractal generation structure is first introduced by FractalNet (Larsson et al., 2016). More recently, Fractal Generative Models (Li et al., 2025) extend the concept of fractals to the domain of image generation.

The Hausdorff dimension proposed by Hausdorff (1918) serves as an important metric for quantifying the complexity of fractal systems, but it is not directly applicable to the field of 3D fractal generation. Fractal3DGen is the first to introduce the concept of normalized Hausdorff dimension, experimentally demonstrating that Fractal3DGen exhibits non-integer dimensional characteristics.

### 2.2 3D AUTOREGRESSIVE GENERATION

Autoregressive (AR) models have seen rapid advances in deep learning and are widely used for 3D generation. Mesh-based AR models (Nash et al., 2020; Chen et al., 2024a;b; Hao et al., 2024) struggle to preserve geometric details and topological stability, while implicit representation methods (Jun & Nichol, 2023; Mittal et al., 2022; Chen et al., 2025) require extensive point sampling, leading to high computational costs. Point cloud and voxel-based AR models (Mo et al., 2019; Cheng et al., 2022; Wei et al., 2025) are simpler but face trade-offs between resolution and memory.

Fractal3DGen is a voxel-based autoregressive model that progressively generates voxels in a fractal manner, preserving geometric details while reducing model complexity. By reusing a single fractal generator in a divide-and-conquer approach, it achieves a more concise and generalizable design. An adaptive pruning mechanism further improves efficiency by skipping empty voxels.

### 2.3 3D GENERATION

Currently, there are three main technical approaches for 3D generation. The first approach uses Generative Adversarial Networks (Goodfellow et al., 2014; Chen & Zhang, 2019) to synthesize 3D models (Wu et al., 2016; Zheng et al., 2022). This approach can efficiently generate high-resolution 3D samples, but it often suffers from training instability and mode collapse. The second approach combines the generative approach of Diffusion Models (Zhou et al., 2021; Nichol et al., 2022; Zhang et al., 2023; Cheng et al., 2023; Erkoç et al., 2023; Chou et al., 2023; Gupta et al.,

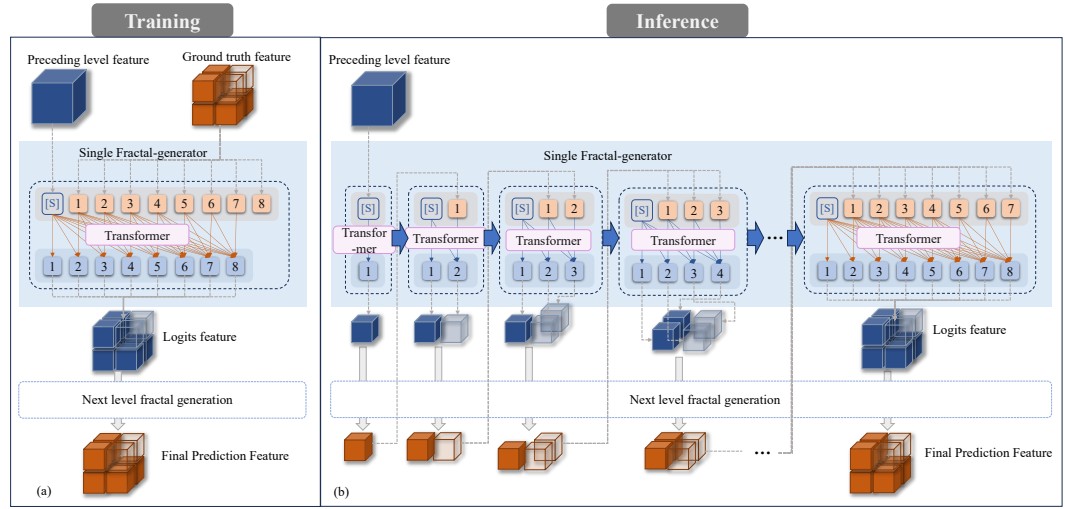

Figure 3: **Autoregressive in Single Fractal-generator.** The fractal-generator inserts conditional embeddings at the head of the sequence to initiate the autoregressive process. As shown in (a), during training, the predicted logits are computed via self-attention based on ground truth. In inference (b), the next token prediction is made conditioned on previously generated logits.

2023; Lan et al., 2024; Wang et al., 2025). This approach excels in generation quality and diversity, yet its iterative denoising process typically leads to high computational costs and slow generation speed. The third approach is 3D autoregressive generation. By decomposing complex 3D structures into manageable steps via sequence modeling, this approach offers promising controllability, though balancing generation quality and sequence length remains a key research challenge. This approach, which this paper explores, is elaborated in Section 2.2.

## 3 METHOD

Fractal3DGen generates voxel representations of 3D objects for a specific category and comprises four key components: (1) a Transformer-based VQ-VAE with encoder $E$ and decoder $D$ that compresses 3D geometry into a low-dimensional latent space; (2) a fractal network in the latent space built from self-similar single fractal-generators for hierarchical refinement; (3) Transformer-based single fractal-generators that assemble the fractal network; and (4) pruning discriminators that determine whether voxel blocks require further refinement. Section 3.1 presents the overall architecture, Section 3.2 describes how single fractal-generators produce logits at arbitrary levels, and Section 3.3 details key design elements, including pruning discriminators and the normalized Hausdorff dimension metric for the 3D fractal structure.

### 3.1 3D FRACTAL ARCHITECTURE

**Sparse Convolution-based VQ-VAE.** We develop a Sparse Convolution-based VQ-VAE (Tang et al., 2022; 2023) to compress 3D voxelized signed distance fields (SDFs) into compact discrete codes. Following the standard VQ-VAE framework, the encoder maps sparse SDF inputs into a latent embedding space, which is then quantized via a learnable codebook. The decoder reconstructs the SDF from these quantized embeddings back to the original resolution. The model is trained with a combination of vector quantization and reconstruction losses:

$$\mathcal{L}_{\text{vae}} = \mathcal{L}_{\text{vq}} + \mathcal{L}_{\text{sdf}}, \tag{1}$$

where $\mathcal{L}_{\text{sdf}}$ is the mean squared error between the reconstructed and ground-truth SDFs. The resulting embeddings form the latent space for 3DFractalGen.

Our encoder employs sparse 3D convolutional residual blocks with downsampling to reduce spatial resolution while increasing feature dimensionality. The decoder mirrors this structure with upsam-

pling and sparse residual blocks to recover fine-grained geometry. This design produces compact discrete representations efficiently while preserving high-fidelity 3D reconstruction.

**3D Fractal Generation.** As shown in Fig. 2, the fractal hierarchical construction operates entirely in latent space, where autoregressive modules at multiple scales enable progressive coarse-to-fine detail generation. Fractal3DGen extends the Fractal Generative Models architecture by incorporating a MAR (Li et al., 2024) variant with self-attention in each module, capturing both local and global dependencies. It further adopts the RoPE3D positional encoding from OctGPT (Wei et al., 2025). The structured hierarchy of fractal generators allows each module to process features from preceding levels, refine them, and propagate the results downstream, supporting progressive hierarchical refinement.

Formally, let $l = 1, 2, \ldots$ denote the stratified level index. At hierarchy level $l$, the transformer output of the $i^{\text{th}}$ fractal-generator is expressed as:

$$(y^{(l)}, m^{(l)})_i = \mathrm{F}_i^{(l)}(y_i^{(l-1)}). \tag{2}$$

In this framework, $\mathrm{F}_i^{(l)}$ represents the $i^{\text{th}}$ fractal-generator at level $l$, serving as the autoregressive module that performs fine-grained fractal generation on the $i^{\text{th}}$ voxel block from the previous hierarchy. Its input, $y_i^{(l-1)}$, is the $i^{\text{th}}$ output from the preceding level. After processing, the module outputs $y^{(l)}$, refining the previous results into $y^{(l)} = (y_1^{(l)}, y_2^{(l)}, \ldots, y_N^{(l)})$, and $m^{(l)}$, representing the pruning decision for the current level (detailed in Section 3.3).

We provide additional multi-resolution visualizations in Appendix A.1, which illustrate how the fractal generator progressively refines the same 3D shape from coarse global structure to high-frequency local details.

## 3.2 Single Fractal-generator

The core fractal architecture of Fractal3DGen follows a divide-and-conquer strategy, realized through the fundamental Single Fractal-generator module. Each fractal-generator receives input from voxel blocks of the preceding level and uses these coarse features to generate multiple refined sub-voxel features, progressively increasing resolution. At the final level, the fractal-generator predicts voxel-wise features containing detailed model information, which are then converted into high-resolution voxel occupancy by the VQ-VAE decoder.

Distinct traversal strategies are employed to effectively utilize these fractal-generators during training and inference. During training, the architecture is traversed in a breadth-first order (Fig. 3 (a)), whereas inference follows a depth-first traversal (Fig. 3 (b.1) & (b.2)), recursively propagating generated outputs to ensure coarse-to-fine autoregressive generation.

For instance, in 3D generation tasks with a latent space resolution of $64^3$, the input condition is a class label encoding the complete information of the $64^3$ voxel block. At level 0, the label embedding is fed into the fractal generator to represent this large voxel block. The output is a token sequence of size $8 \times 8 \times 8$ (512 tokens), corresponding to 512 voxel blocks obtained by subdividing the $64^3$ volume into $8^3$ sub-blocks. At the final level, the output $y^{(l)}$ reduces to single-voxel predictions. The final output is then optimized using a per-voxel cross-entropy loss:

$$\mathcal{L}_{\text{voxel}} = -\frac{1}{N} \sum_{i=1}^{N} \Big[ \alpha \hat{v}_i \log \sigma(v_i) + (1-\alpha)(1-\hat{v}_i) \log (1 - \sigma(v_i)) \Big], \tag{3}$$

where $\hat{v}_i$ is the ground-truth voxel feature, $v_i \in \mathbb{R}$ is the predicted logits for voxel occupancy, $\sigma(v_i) = \frac{1}{1+e^{-v_i}}$ is the sigmoid activation function, and $\alpha \in [0, 1]$ controls the positive class weight.

## 3.3 Pruning and Normalized Hausdorff Dimension

Voxelized 3D models contain many unoccupied voxels that often form clusters. Leveraging this, Fractal3DGen employs a pruning operation to skip generation in empty voxel blocks, shown in Fig. 4. The pruning mask $m^{(l)}$ in Equation (2) decides whether to proceed to the next fractal level, significantly improving efficiency.

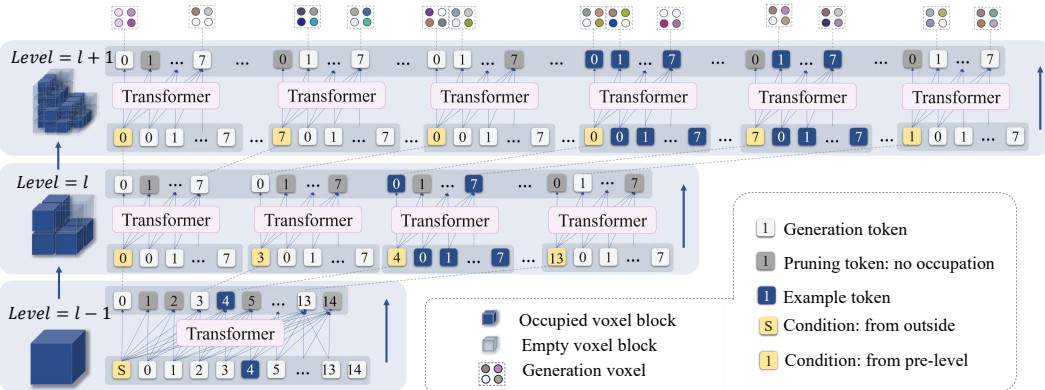

Figure 4: **Pruning in Fractal3DGen.** Dark blue tokens illustrate example tokens in the coarse-to-fine voxel block generation (left). Gray tokens denote empty voxel blocks, which are pruned without further fractal generation. White tokens represent occupied blocks, activating sub-modules for next-level generation and serving as conditional inputs (yellow tokens) for subsequent levels. First-level conditions are derived from external embeddings, and the final level produces the generated voxels.

The pruning mask $m \in \{0, 1\}^n$ records binary refinement decisions: 1 indicates occupied blocks requiring further generation, while 0 indicates empty blocks to prune. Specifically, the output $y \in \mathbb{R}^{N \times d}$ from the Single Fractal-generator is passed through a fully connected layer to produce $m = (m_1, \ldots, m_N)$, where $m_i = 0$ corresponds to gray tokens in Fig. 4 (pruned) and $m_i = 1$ corresponds to white tokens (occupied), which then serve as conditional inputs (yellow tokens) for the next level. The dark-blue "example token" in Fig. 4 serves only as a visual illustration aligned with the coarse-to-fine voxel block shown on the left. It has no semantic or functional difference from the other gray tokens and does not represent any additional token type in our model.

Pruning is optimized via binary cross-entropy loss:

$$\mathcal{L}_{\text{pruning}} = -\frac{1}{N} \sum_{i=1}^{N} \Big[ \alpha \hat{m}_i \log \sigma(m_i) + (1 - \alpha)(1 - \hat{m}_i) \log (1 - \sigma(m_i)) \Big], \tag{4}$$

where $\hat{m}_i \in \{0, 1\}$ is the ground-truth occupancy, $m_i$ is the predicted logit, $\sigma$ is the sigmoid, and $\alpha$ weights the positive class. The overall training objective combines voxel and pruning losses:

$$L = \alpha \mathcal{L}_{\text{voxel}} + (1 - \alpha) \sum_{l=1}^{l_{\text{num}}} \mathcal{L}_{\text{pruning}}^{(l)}, \tag{5}$$

where $l_{\text{num}}$ is the number of levels and $\alpha$ controls the weighting.

To evaluate pruning and capture the fractal nature of the generation, we define the Normalized Hausdorff Dimension. Let $R^{(l)}$ and $R^{(l+1)}$ be the resolutions of levels $l$ and $l + 1$, with scaling ratio $\epsilon^{(l)} = \frac{R^{(l)}}{R^{(l+1)}}$. The Hausdorff dimension of the $i^{\text{th}}$ Fractal-generator at level $l$ is

$$d_{H\,i}^{(l)} = \log_{\epsilon^{(l)}} n_i^{(l)}, \tag{6}$$

where $n_i^{(l)}$ is the number of tokens propagated after pruning. The normalized dimension at level $l$ is and for the entire process,

$$d_H = \frac{1}{l_{\text{num}}} \sum_{l=1}^{l_{\text{num}}} \frac{1}{n^{(l-1)}} \sum_{i=1}^{n^{(l-1)}} d_{H\,i}^{(l)}, \quad 1 < d_{H\,i}^{(l)} \le 3. \tag{7}$$

A detailed derivation is provided in Appendix A.2. Using a two-level FractalGen network to generate 300 car models, we observed a maximum $d_H$ of 2.494 and an average of 2.272, showing that Fractal3DGen adapts dynamically while remaining below 3.0. In contrast, the latest 2D fractal

generation model(Li et al., 2025) maintains a fixed Hausdorff dimension of 2.0, limiting its representational capacity and computational efficiency.

## 4 IMPLEMENTATION DETAILS

This section provides a overview of the VQ-VAE training details and the 3D fractal configuration in the latent space.

**VQ-VAE Training Details:** The VQ-VAE received inputs of resolution $256^3$ and encoded them into a latent space of $64^3$. A lightweight codebook with a vocabulary size and embedding dimension of 32 was used, and the decoder reconstructed latent features back to $256^3$. The VQ-VAE was trained for 1000 epochs with a batch size of 4 using the AdamW optimizer (Loshchilov & Hutter, 2019) (learning rate $5 \times 10^{-4}$, gradient clipping norm 1.0).

**3D Fractal Configuration:** In the latent space, Fractal3DGen uses the stochastic-order autoregressive scheme from MAR, with an input resolution of $64^3$ and a default two-level fractal architecture. We choose two levels because, at the target resolution of $256^3$, adding more hierarchical levels does not necessarily improve efficiency or quality. Deeper hierarchies introduce extra autoregressive overhead and higher memory-transfer cost, and overly fine subdivision may cause error accumulation and unwanted structural deformation. Therefore, the number of fractal levels should be chosen carefully instead of increased blindly. In our experiments, the two-level design provides the best balance among generation quality, speed, and GPU memory usage: it captures global structure well while avoiding unnecessary computational cost. A deeper design (such as five levels) mainly becomes beneficial only when scaling to much higher resolutions, which we plan to explore in future work. The two-level fractal autoregressive module is trained for 1000 epochs using AdamW (learning rate $1 \times 10^{-4}$, batch size 4, weight decay $1 \times 10^{-4}$), and all experiments are conducted on NVIDIA H20 GPUs.

## 5 EXPERIMENTS

This section first compares Fractal3DGen with other 3D shape generative models, and then presents the visualization of Fractal3DGen's 3D generation quality in Section 5.1. Finally, the ablation studies and application analysis are discussed in Sections 5.2 and 5.3, respectively.

### 5.1 QUANTITATIVE EVALUATION OF GENERATION PERFORMANCE

This subsection evaluates Fractal3DGen on the ShapeNet dataset (Chang et al., 2015) and compares it with existing state-of-the-art 3D generation methods. We benchmark against GAN-based methods (IM-GAN (Chen & Zhang, 2019), SDF-StyleGAN (Zheng et al., 2022)), diffusion-based methods (Wavelet-Diffusion (Hui et al., 2022), MeshDiffusion (Liu et al., 2023), SPAGHETTI (Hertz et al., 2022), LAS-Diffusion (Zheng et al., 2023), XCube (Ren et al., 2024), OctFusion (Xiong et al., 2025), 3DShape2VecSet (Zhang et al., 2023)), and autoregressive methods (MeshGPT (Siddiqui et al., 2024), 3DILG (Zhang et al., 2022), OctGPT (Wei et al., 2025)), the latter being most closely related to our approach. MeshGPT is mesh-based and 3DILG encodes 3D shapes as neural fields, differing from our voxel-based 3D point cloud representation. While OctGPT also generates point clouds, it uses an octree structure rather than directly processing voxels. Interactive3D (Dong et al., 2024) and CoPart (Dong et al., 2025) focus on interactive and part-based generation, respectively, addressing different problems and thus are excluded from our comparison.

**Quantitative evaluation.** Quantitative FID results are summarized in Tab. 1. Since some model checkpoints are unavailable, OctGPT's FID metrics are used as the benchmark for fair comparison. Following OctGPT's training protocol, Wavelet-Diffusion was trained on three categories, and SPAGHETTI and MeshGPT on two. Training datasets for MeshGPT, 3DILG, and 3DShape2VecSet also slightly differ from others. The results for category-wise and joint training are presented in the upper and lower sections of the table, respectively.

Fractal3DGen achieves the best performance, surpassing all baselines in average FID across five categories. It outperforms OctGPT, the previous top autoregressive model, in every category, with

Table 1: **The quantitative comparison of shading-image-based FID.** Bold numbers indicate the best metrics, underlined numbers the second-best. Shaded rows denote autoregressive models, and blue numbers highlight Fractal3DGen's FID improvement over the previous autoregressive models.

| Method | Car | Chair | Airplane | Table | Rifle | Average |
|---|---|---|---|---|---|---|
| IM-GAN | 141.2 | 63.42 | 74.57 | 51.70 | 103.3 | 86.84 |
| SDF-StyleGAN | 97.99 | 36.48 | 65.77 | 39.03 | 64.86 | 60.83 |
| Wavelet-Diffusion | N/A | 28.64 | 35.05 | 30.27 | N/A | N/A |
| MeshDiffusion | 156.21 | 49.01 | 97.81 | 49.71 | 87.96 | 88.14 |
| SPAGHETTI | N/A | 65.26 | 59.21 | N/A | N/A | N/A |
| LAS-Diffusion | 80.55 | 20.45 | 32.71 | _17.25_ | 44.93 | 39.18 |
| XCube | 80.00 | _18.07_ | **19.08** | N/A | N/A | N/A |
| OctFusion | 78.00 | **16.15** | _24.29_ | **17.19** | 30.56 | 33.24 |
| MeshGPT | N/A | 37.05 | N/A | 25.25 | N/A | N/A |
| OctGPT | _64.45_ | 31.05 | 27.47 | 19.64 | _21.91_ | _32.90_ |
| Fractal3DGen | **46.89**↓17.56 | 26.60↓4.45 | 25.44↓2.03 | 23.87 | **16.47**↓5.44 | **27.85**↓5.05 |
| 3DShape2VecSet | 110.12 | 21.21 | 46.27 | 25.15 | 54.20 | 51.39 |
| LAS-Diffusion | 86.34 | 21.55 | 43.08 | _17.41_ | 70.39 | 47.75 |
| OctFusion | 80.97 | **19.63** | 30.92 | 17.49 | _28.59_ | 35.52 |
| 3DILG | 164.15 | 31.64 | 54.38 | 54.13 | 77.74 | 76.41 |
| OctGPT | _62.40_ | 28.28 | _29.27_ | 20.64 | **27.21** | _33.56_ |
| Fractal3DGen | **47.27**↓15.13 | _20.96_↓7.32 | **28.25**↓1.02 | **17.28**↓3.36 | 28.67 | **28.49**↓5.07 |

a notable FID reduction of 17.56 on the car category. Under category-wise training, Fractal3DGen achieves state-of-the-art performance among AR models on four categories and surpasses diffusion-based methods on two others. Remarkably, under joint training, it exceeds diffusion-based methods on three categories. To verify the efficiency, we generated 100 cars on an H20 GPU using both FractalGen and OctGPT. FractalGen achieves a 1.85× speedup, reducing the average inference time per shape from 54.1 s to 29.3 s.

**Qualitative Analysis.** Qualitative results in Fig. 5 demonstrate that our method generates shapes with superior detail compared to existing approaches. This improvement stems from our autoregressive fractal generation framework, which leverages a coarse-to-fine, multi-resolution strategy. In contrast, GAN-based methods often suffer from mode collapse and blurred fine features, diffusion-based methods require cascaded stages with potential error accumulation, and prior autoregressive models struggle to maintain structural coherence while capturing intricate details. By recursively partitioning voxel models into smaller sub-models, Fractal3DGen efficiently models both macroscopic and microscopic structures, yielding higher-quality shapes with enhanced visual fidelity.

## 5.2 Ablation Study and Discussions

Ablation studies were conducted to assess the influence of fractal depth and pruning strategies. To enable the inclusion of a larger number of Fractal generators within the limitations of GPU memory, the number of Transformer Blocks was reduced in the ablation setting, with further implementation. The evaluation was subsequently carried out on the car category to validate the effectiveness of the analysis.

**Ablation study on the fractal architecture.** The number of hierarchical levels was varied from 1 to 4 to evaluate shape generation quality. When the number of levels is 1, the model loses its fractal structure, and the Transformer sequence length reaches $64^3$, exceeding memory limits, indicating that autoregressive modeling of such long sequences is infeasible without a fractal design. As shown in Fig. 6 (a), ablation results start from Level 2. Performance improves with increasing levels: 3 levels capture global structure and fine details better than 2, while 4 levels achieve the best results. These findings confirm the effectiveness of fractal hierarchy in balancing global structure and fine-grained detail.

**Ablation study on pruning.** We validated the pruning time efficiency on a five-level FractalGen model. The horizontal axis represents the average shape generation time after pruning the first $n$

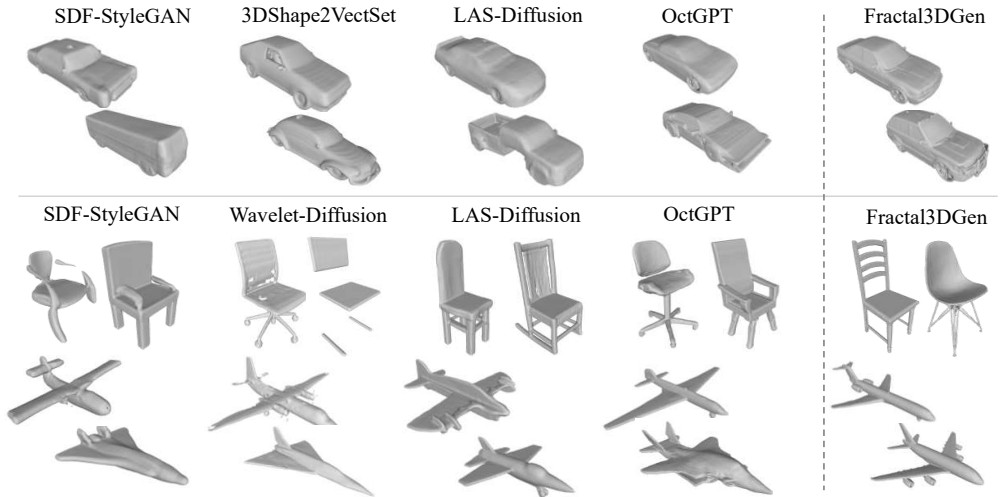

Figure 5: **Qualitative comparison results.**

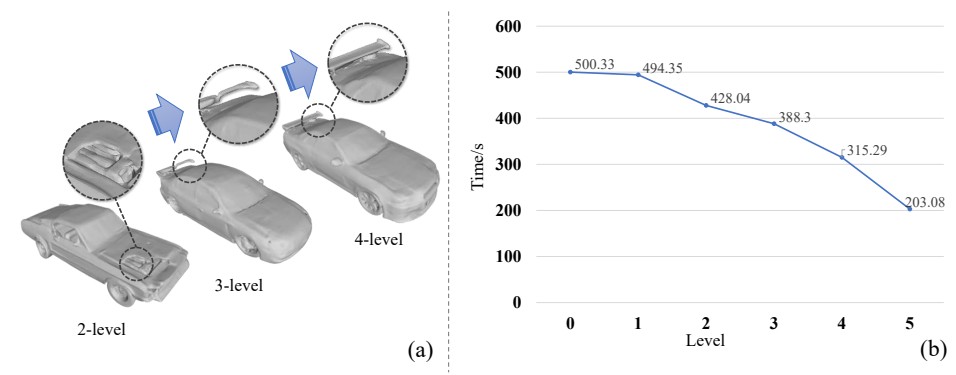

Figure 6: **Ablation Results.**

level. It can be observed that, as the pruning strategy is applied more extensively, the model's generation time decreases significantly, which is attributed to the sparsity characteristics of 3D objects. Overall, the experimental results shown in Fig. 6 (b) demonstrate the effectiveness of the fractal architecture in improving model efficiency.

### 5.3 APPLICATIONS AND DISCUSSIONS

In this section, we showcase several practical applications enabled by Fractal3DGen, highlighting its capability to generate high-quality 3D shapes from diverse input modalities.

**Text-to-3D Generation.** By leveraging CLIP (Radford et al., 2021) for cross-modal alignment, Fractal3DGen can generate 3D shapes from language descriptions. Trained on the Text2Shape dataset, our model produces 3D shapes that are consistent with both the semantic content and structural details of the input text (Fig. 7 (a)), facilitating effective cross-modal 3D content creation.

**Image/Sketch-to-3D Generation.** Fractal3DGen can generate 3D shapes conditioned on images or sketches. Input images from ShapeNet (Chang et al., 2015) and sketches prepared following LAS-Diffusion (Zheng et al., 2023) are encoded using DinoV2, and the model produces 3D shapes that closely preserve the structure and details of the inputs (Fig. 7 (b)).

More results are presented in Appendix A.3.

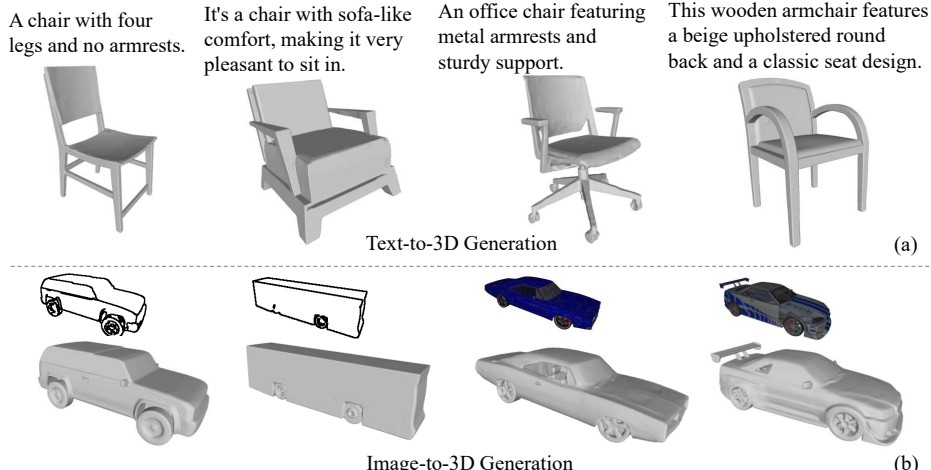

Text-to-3D Generation (a)

Image-to-3D Generation (b)

Figure 7: **Qualitative results of practical applications.**

## 6 CONCLUSIONS

We present Fractal3DGen, a novel fractal-autoregressive model for high-quality and efficient 3D shape generation. By integrating fractal theory with transformer-based autoregressive modeling, our approach achieves a better trade-off between generation quality and computational efficiency than previous methods. The key innovations comprise a hierarchical fractal generation architecture that reuses self-similar transformers across scales, and an adaptive pruning mechanism that skips empty voxel regions to accelerate inference. Experimental results on the ShapeNet benchmark demonstrate state-of-the-art performance, outperforming other existing methods.

For future work, we plan to extend Fractal3DGen to other 3D representations such as point clouds and meshes, explore conditional generation capabilities for more controllable content creation, and further optimize the model for real-time applications.

## ETHICS STATEMENT

This work strictly adheres to the ICLR Code of Ethics. All experiments were conducted using publicly available datasets (e.g., ShapeNet), which do not contain human subjects, personal or sensitive information. Our research does not involve discrimination, bias, or unfair practices, nor does it pose foreseeable risks of harmful misuse. The proposed method is intended solely for legitimate applications such as academic research and industrial design in 3D modeling and generation. All experiments follow research integrity principles, and the reported results are truthful and reproducible. The authors declare no conflicts of interest or inappropriate sponsorship.

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

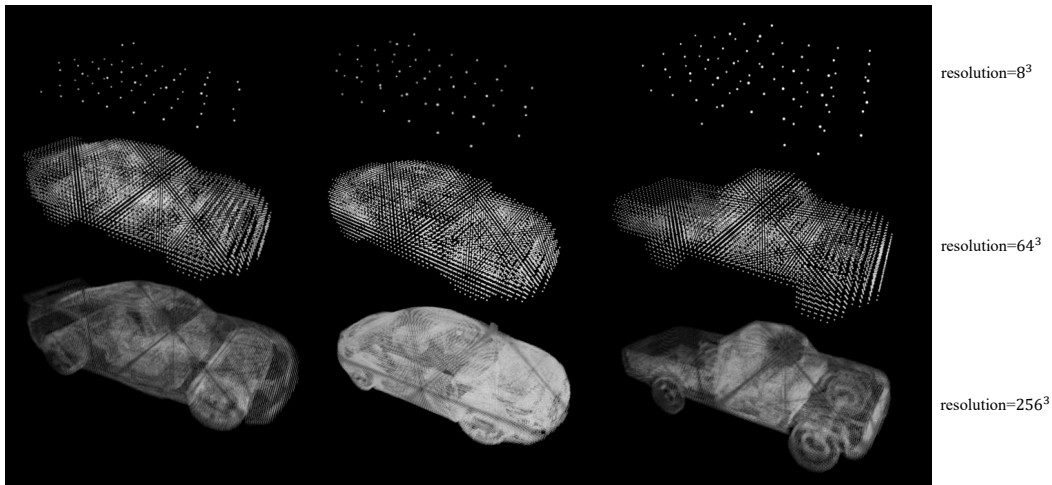

resolution=$8^3$

resolution=$64^3$

resolution=$256^3$

Figure 8: **Multi-Resolution Visualization of the Fractal3DGen Coarse-to-Fine Generation Process.**

# A APPENDIX

## A.1 INTERPRETABILITY OF FRACTAL3DGEN VIA COARSE-TO-FINE FRACTAL GENERATION

Fig. 8 illustrates how the same 3D object is generated step by step as the resolution increases from $8^3 \rightarrow 64^3 \rightarrow 256^3$. This process clearly demonstrates how Fractal3DGen refines shapes in a coarse-to-fine manner:

- **Low-resolution level** ($8^3$)**:** At this stage, the model only needs to capture the global structure of the object. The fractal-generator predicts large voxel blocks, allowing the rough silhouette of the car to appear.
- **Mid-resolution level** ($64^3$)**:** When the resolution increases, each coarse block is split into smaller sub-blocks. The same fractal-generator is reused to refine these sub-blocks using features from the previous level. Details such as roof curvature and window positions start to emerge.
- **High-resolution level** ($256^3$)**:** At the final level, the model produces voxel-wise features and reconstructs high-frequency geometry using the VQ-VAE decoder. Fine details—including wheel boundaries, headlights, and door seams—are added at this step.

These visualizations show that Fractal3DGen maintains the global structure across levels while gradually adding fine-grained details through recursive refinement, leading to a natural and interpretable generation process.

## A.2 NORMALIZED HAUSDORFF DIMENSION ANALYSIS OF FRACTAL3DGEN

Here, we analyze the complexity of Fractal3DGen through the lens of the normalized Hausdorff dimension. Before pruning, the sequence length $N^l$ of level $(l)$ is given by:

$$N^{(l)} = \left( \frac{R^{(l)}}{R^{(l+1)}} \right)^3 . \tag{8}$$

Due to the pruning mechanism, $0 < n^{(l)} \leq N^{(l)}$, where equality holds if and only if no pruning is applied at this level. When no pruning is applied at a certain level $l$, its normalized Hausdorff dimension can be expressed as:

$$d_{H_i}^{(l)} = \log_{\epsilon^{(l)}} n_i^{(l)} = \log_{R^{(l)}/R^{(l+1)}} \left( \left( \frac{R^{(l)}}{R^{(l+1)}} \right)^3 \right) = 3. \tag{9}$$

When the number of autoregressive modules $n^l$ approaches zero for all levels, $d_{H_i}^{(l)}$ will be:

$$d_{H_i}^{(l)} = \log_{\epsilon^{(l)}} 0 = 1. \tag{10}$$

Thus, the range of the normalized Hausdorff dimension is:

$$1 < d_{H_i}^{(l)} \le 3. \tag{11}$$

Furthermore, it can be derived that the normalized Hausdorff dimension for the entire generation process is:

$$1 < d_H \le 3. \tag{12}$$

### A.3 ADDITIONAL RESULTS FOR APPLICATIONS

For text-conditioned 3D generation, we employ the CLIP ViT-L/14 encoder to extract high-level semantic representations from the input text. We use the pooled global text embedding from the final transformer layer as the conditioning signal for our generator, providing a compact yet expressive semantic descriptor that plays a similar role to the category embedding used in the class-conditioned setting.

For image-conditioned 3D generation, we adopt the DINOv2-Large encoder and use the final-layer [CLS] token as the global image representation. The [CLS] token is chosen because DINOv2's self-supervised training objective explicitly optimizes it to aggregate global semantic information, making it a stable and effective descriptor for downstream tasks such as 3D generation.

Across all modalities, the FractalGen backbone remains unchanged. The fractal hierarchy, network architecture, and structural hyperparameters are identical to those used for category-conditioned generation. Our design is inherently condition-agnostic: the generator requires only a single global conditioning vector, regardless of the modality from which it is derived. Thus, text-based, image-based, and category-based generation differ only in the source of this conditioning vector, without necessitating any modification to the generative architecture. Additional results are presented in Fig. 9.

### A.4 LARGE LANGUAGE MODEL USAGE

In the preparation of this manuscript, the authors employed a large language model (LLM), specifically deepseek, solely as a tool to aid and polish the writing of the English language. The use of the LLM was strictly limited to the post-ideation and post-analysis phases of the research process.

The model was used for the following specific purposes:

- Grammar and Syntax Correction: To identify and correct grammatical errors, typos, and awkward phrasing in text initially drafted by the authors.
- Sentence Rephrasing and Clarity Enhancement: To suggest alternative phrasing for sentences to improve clarity, conciseness, and overall readability, while strictly preserving the original technical meaning and scientific intent.
- Tone and Consistency Check: To ensure a consistent and formal academic tone throughout the manuscript.

It is crucial to emphasize that the core scientific ideas, hypotheses, research design, methodological development, data analysis, interpretation of results, and the drawing of conclusions originated solely from the human authors. All factual claims, citations, and data presented in the paper are the sole responsibility of the authors. The LLM was not used to generate, fabricate, or interpret any scientific data, nor was it used to conduct literature reviews or to generate original scientific concepts. The authors reviewed, edited, and take full responsibility for all content generated by the LLM, ensuring its accuracy and alignment with the authors' intended meaning.

The LLM was not used in any capacity that would qualify it as a contributor to the intellectual content of this work. Therefore, in accordance with the conference/journal policy, the LLM is not listed as an author.

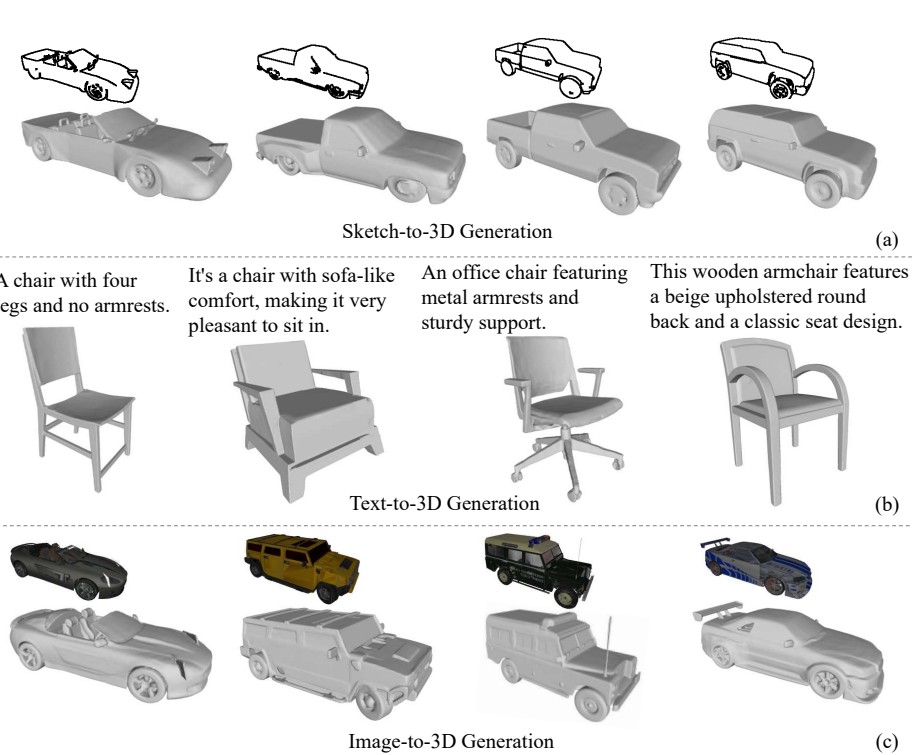

Sketch-to-3D Generation (a)

A chair with four legs and no armrests.

It's a chair with sofa-like comfort, making it very pleasant to sit in.

An office chair featuring metal armrests and sturdy support.

This wooden armchair features a beige upholstered round back and a classic seat design.

Text-to-3D Generation (b)

Image-to-3D Generation (c)

Figure 9: **More qualitative results of practical applications.**

