# OpenReview forum: "FRACTAL3DGEN: COARSE-TO-FINE 3D GENERATION VIA FRACTAL HIERARCHICAL TRANSFORMERS"
_ICLR.cc/2026/Conference — Submitted to ICLR 2026_

### Official Review · Reviewer_R1MH · 2025-10-21

**Soundness:** 3
**Presentation:** 3
**Contribution:** 3
**Rating:** 6
**Confidence:** 4

**Summary:**

The authors propose Fractal3DGen, a hierarchical fractalautoregressive framework that leverages self-similarity across multiple scales for
3D shape generation. Besides, they introduce an adaptive pruning mechanism to improve the efficiency. The experimental results on ShapeNet demonstrate the effectiveness of the proposed approach.

**Strengths:**

1. A new auto-regressive 3D shape generation model with different generative order from OctGPT,  improving both the quality and efficiency of autoregressive models.
2. A 3D fractal pruning strategy for efficient inference.
3. It achieves significant improvements in generation speed, quantitative metrics, and computational efficiency.

**Weaknesses:**

1. The comparison is only conducted on ShapeNet, while OctGPT is also evaluated on Objaverse and Synthetic Rooms dataset. I think ShapeNet is too simple and may not fully reveal the performance of the proposed approach.
2. I think "BIOLOGICAL INSPIRATION OF FRACTAL3DGEN" of Appendix A is over-claimed. It is a new design of generative order built upon OctGPT, and the "biological inspiration" is too vague and not really reflect the contribution.

**Questions:**

N.A.

---

> ### Author Response · Authors · 2025-11-21
>
> **We thank the reviewer for the constructive and insightful feedback**. We sincerely appreciate the time and effort spent evaluating our work and for highlighting both its strengths and the areas where clarification or improvement is needed. Below we address the raised concerns in detail.
>
> **1. Regarding the evaluation on ShapeNet only**
>
> Thank you for pointing out the limitation of evaluating our method solely on the ShapeNet dataset. We fully agree that broader evaluations are beneficial for demonstrating generality.
> We chose to focus on the ShapeNet dataset mainly to **stay consistent with the existing baselines**. Most current methods are also evaluated only on ShapeNet. For example, OctFusion, MeshGPT, 3DILG, LAS-Diffusion, 3DShape2VecSet, SPAGHETTI, MeshDiffusion, Wavelet-Diffusion, Sdf-StyleGAN, and IM-GAN all use only ShapeNet. We currently focus on completing fair and consistent evaluations on ShapeNet. Experiments on other datasets are already in progress, and we will gradually include them in our future work.
>
> **2. Regarding “Biological Inspiration of Fractal3DGen” (Appendix A)**
>
> We sincerely thank the reviewer for this thoughtful and insightful comment. Your observation made us realize that the description in this part was not sufficiently precise. Following your suggestion, we revised the appendix accordingly and remove the ‘BIOLOGICAL INSPIRATION OF FRACTAL3DGEN’ section to avoid potential misunderstandings.
>
> **Once again, we appreciate the reviewer’s thoughtful feedback.**

---

### Official Review · Reviewer_chL5 · 2025-10-23

**Soundness:** 3
**Presentation:** 3
**Contribution:** 3
**Rating:** 4
**Confidence:** 3

**Summary:**

The paper presents a 3D generation method inspired by fractal formulation. The sparse 3D volume hierarchically upsamples to finer resolution, with an additional pruning framework that maintains the underlying sparsity and efficiency. They design the order of traversal for training and inference, and propose a normalized Hausdorff dimension to apply loss in different resolutions. The results demonstrate superior performance against other baselines with autoregressive generation methods. In addition to the quality of the results, the hierarchical generation is fast and efficient. Additional results include ablation on the number of layers and various application scenarios.

**Strengths:**

The paper builds on the fractal generation method in 2D and applies a similar technique in 3D. To transfer to the 3D domain, the representation incorporates a sparse voxel representation, which is also widely used in 3D volumetric representation. But the pruning mechanism and the loss formulation with binary masks are tailored to the representation. Additionally, the proposed normalized Hausdorff dimension provides necessary scaling for different resolutions, considering the subdivision, which nicely integrates the proposed method.

**Weaknesses:**

- The effect of Hausdorff dimension needs further evaluation to be perceived as a valid contribution.

- It seems like a two-layer architecture is used for most of the results (stated in Section 4.1), which might not be challenging to consider as "fractal". Fractals often imply a greater number of recursive formulas, or almost infinite layers of progression.

- The results are evaluated in the ShapeNet dataset, which is relatively old and small in size. Also, the amount of improvement is not very significant, and even underperforms many existing works, depending on the shape categories.

**Questions:**

- Figure 3 is hard to understand, which should be critical to improve the clarity of the exposition.

- Why do you propose a different traversal for training (breadth-first) and inference (depth-first)?

- What are "example tokens" in Figure 4?

- Is layer and level the same thing in Section 4.3? Why not use five-level architecture if they are more accurate and faster?

**Details Of Ethics Concerns:**

While generative models may raise ethics concerns, the proposed work deals with ShapeNet, which is a relatively narrow range of objects that are already widely used, and does not include any real data or privacy concerns.

---

> ### Author Response · Authors · 2025-11-21
>
> We sincerely thank the reviewer for the thoughtful and constructive comments. We address each of the concerns in detail below and incorporate the suggested clarifications and improvements in the revised version.
>
> **Weaknesses**
>
> **1. “The effect of Hausdorff dimension needs further evaluation to be perceived as a valid contribution.”**
>
> Thank you for raising this point. Since this work is the first to introduce fractal generation into the 3D domain, future studies will likely offer additional baselines for broader comparison, which can further validate and extend the effectiveness of the Normalized Hausdorff Dimension ($d_H$).
> In our current work, the role of $d_H$ is mainly reflected in the following two aspects:
>
> **Objective verification of fractal generation.**
>
> The key innovation of Fractal3DGen lies in its multi-scale self-similar generation process. $d_H$ directly measures the fractal complexity of the generated structures ($1 < d_H < 3$). In our experiments, it remains stable with an average value of 2.272, showing that the model learns fractal-like structures rather than simple voxel filling.
>
> **A quantitative indicator for pruning effectiveness.**
>
> Because effective pruning reduces the number of tokens that continue through the multi-scale generation process, the normalized Hausdorff dimension also decreases, making it a direct quantitative indicator of pruning efficiency. For example, in Fractal3DGen, this dimension stays within the reasonable range of 1–3 and is noticeably lower than the no-pruning upper bound of 3.0. This aligns with the model’s 77.6% spatial pruning efficiency and 1.85× inference speedup on ShapeNet, showing a clear connection between the dimension measure and the actual efficiency gains.
>
> In summary, $d_H$ serves as a core component for validating fractal generation behavior and assessing pruning strategies, rather than an auxiliary metric. We improve the discussion of the role and impact of the normalized Hausdorff dimension in the final version.
>
> **2. “Only two layers are used, which might not be sufficiently ‘fractal’.”**
>
> We appreciate this insightful comment and provide further clarification below:
>
> **On the meaning of “Fractal” in our model**
>
> Our use of the term “Fractal” does not refer to infinite mathematical recursion. Instead, it highlights the self-similar and recursive modular design of our architecture. Similar to FractalNet and other fractal-based models, the key idea is the repeated structure across scales and the hierarchical generation process, rather than the number of levels or infinite depth.
> In 3D generation, true infinite recursion is neither possible nor necessary. Our model uses a finite number of recursive levels to capture global-to-local shape structures while keeping the model scalable and efficient.
>
> **Why two levels are used in our experiments**
>
> The difference between two levels and five levels lies only in the depth of recursion. Each level uses the same fractal-generator unit. More levels mean finer voxel block subdivision and potentially richer details at higher resolutions.
> In this paper, we use $256^3$ SDF inputs and a $64^3$ latent space. Under this setting, a two-level hierarchy is sufficient for coarse-to-fine generation (Section 4.1) and provides the best balance between detail quality and computational cost.
> When the model is extended to higher resolutions (e.g., $512^3$ or $1024^3$), additional levels can be naturally introduced. This demonstrates the scalability and future potential of Fractal3DGen.
>
> We clarify this point more explicitly in the revised version.
>
> **3. “ShapeNet is small and outdated; improvements are not very significant.”**
>
> We chose to focus on the ShapeNet dataset mainly to stay consistent with the existing baselines. Most current methods are also evaluated only on ShapeNet. For example, OctFusion, MeshGPT, 3DILG, LAS-Diffusion, 3DShape2VecSet, SPAGHETTI, MeshDiffusion, Wavelet-Diffusion, Sdf-StyleGAN, and IM-GAN all use only ShapeNet. We currently focus on completing fair and consistent evaluations on ShapeNet. Experiments on other datasets are already in progress, and we will gradually include them in our future work.
>
> We emphasize two additional points:
>
> **Quality**
>
> Although performance varies across shape categories, the average quality across metrics is competitive or better than diffusion-based methods.
>
> **Efficiency advantage**
>
> Fractal3DGen shows a clear and significant efficiency improvement:
>
> Fractal3DGen: 29.3 seconds per sample
>
> OctGPT: 54.1 seconds per sample
>
> $→ 1.85×$ faster generation

---

> ### Author Response · Authors · 2025-11-21
>
> **Questions:**
>
> **1. “Figure 3 is hard to understand.”**
>
> Thank you for pointing this out. We update Figure 3 in the revised version to improve clarity.
>
> **2. “Why use breadth-first traversal for training and depth-first for inference?”**
>
> We updated Figure 3 in the paper to better illustrate the behavior of the Single Fractal-generator during training and inference.
> During training, we use a breadth-first traversal because the model can **take advantage of the full ground-truth (GT) structure through teacher forcing**. This means the model can directly access the real features from both upper and lower levels, allowing all voxel blocks at the same level to be trained in parallel. This leads to stable and efficient supervised learning, as shown in Figure 3(a).
>
> In contrast, during inference, there is no ground truth available. The model must follow the real autoregressive generation process: the features of the current voxel block **must be fully generated before they can be used to produce the finer-level structures**. As shown in Figure 3(b), this naturally becomes a top-down, step-by-step expansion, which corresponds to a depth-first traversal (DFS). This strategy ensures that the coarse-to-fine fractal generation is unfolded correctly and maintains the causality required during inference.
>
> **3. “What are ‘example tokens’ in Figure 4?”**
>
> The “example token” marked in Figure 4 is not different in meaning or function from the other gray tokens. It is highlighted in dark blue only to **match the dark-blue voxel block shown on the left, serving as an example to illustrate our coarse-to-fine generation process**. In other words, this token is used purely for visualization and does not represent any additional or special type of token in our model. Thank you very much for your comment—we have updated the main text of the paper to make this explanation clearer.
>
>
> **4. “Are layer and level the same in Section 4.3? Why not use five levels if faster and more accurate?”**
>
> Yes and we will unify “layer” and “level” to avoid ambiguity.
> Thank you for highlighting this terminology issue.
>
> Regarding why we do not adopt the five-level architecture in final experiments: More hierarchical levels do not necessarily improve efficiency or quality; deeper hierarchies introduce additional autoregressive overhead and memory transfer costs, while overly fine subdivision may cause error accumulation and structural deformation. **Thus, selecting the number of levels requires a balance of efficiency and quality, and increasing the depth may not always be beneficial.**. For the $256^3$ setting used in the main paper, a two-level hierarchy achieves the best balance among generation quality, speed, and GPU memory usage. The benefits of a five-level architecture mainly emerge when scaling to much higher resolutions, which we plan to explore in future extensions.
> We provide clearer explanations in the revised version.
>
> We sincerely appreciate the reviewer’s constructive feedback.
> We revise the manuscript to:
>  - improve figure clarity,
>  - unify terminology,
>  - expand explanation of Hausdorff dimension,
>  - provide clearer justification for hierarchical depth choices, and
>  - strengthen the discussion on dataset selection and experimental results.
>
> **Thank you again for your valuable comments.**

---

> > ### Comment · Reviewer_chL5 · 2025-11-27
> >
> > Thank you for the detailed answer.
> >
> > I still consider the presented results to be limited. As the authors suggest, some works still report results on ShapeNet, but other baselines (e.g., OctGPT) include more 3D objects and support scene-level generation. Other works that use ShapeNet alone also typically include additional components, e.g., texture generation (OctFusion) or mesh topology resolution (MeshGPT). I understand the concept of fractals and why the authors used two layers in their setup. But at the same time, the rebuttal implies the rigidity of the proposed method and cannot outperform beyond the presented setup. It would be more potent if having more layers could provide more efficiency or other interesting emergent behavior. It is not clear how the work can be extended beyond heavily-tuned set-up.1~2x efficiency also does not appear as a significant advancement to me.

---

### Official Review · Reviewer_2bCM · 2025-10-30

**Soundness:** 3
**Presentation:** 3
**Contribution:** 3
**Rating:** 4
**Confidence:** 5

**Summary:**

This paper introduces a coarse-to-fine fractal-autoregressive framework for 3D shape generation. Shapes are encoded as sparse SDF voxels into a latent space via VQ-VAE; a self-similar Transformer fractal generator is reused across levels to refine voxel blocks from global structure to local details. Inference uses an adaptive pruning mask to stop refining empty regions. On ShapeNet, the method achieves lower FID than prior baselines and ~1.85× faster inference than OctGPT, while preserving fine geometry. The method further supports text- and image-conditioned generation.

**Strengths:**

1.The paper has a conceptually elegant framework, which proposes a dynamic, recursive generation process. This "divide-and-conquer" strategy is a principled approach to managing the high dimensionality of 3D data.

2.This work achieves state-of-the-art performance on the ShapeNet benchmark,with an impressive speedup in average inference time by pruning.

3.The paper defines Normalized Hausdorff Dimension with detailed derivation to clearly evaluate the computational efficiency.

**Weaknesses:**

1. Ambiguity in novelty.The hierarchical structure and autoregressive fractal generators are similar in “Fractal Generative Models”,while the VQ-VAE encoder is used in OctGPT.

2.Experimental results in Table 1 only use FID as metric, some others can be introduced.Also,some baseline models like MeshGPT are not trained on all the categories.The superiority of Fractal3DGen will be more convincing if all categories are evaluated.

3.The model details in practical applications like text- and image-to-3D generation using CLIP or DINOv2 encoders are not clear.

Shapenet is a pretty easy dataset, which can be easily overfitting. Shall it be reasonable (and make sense) to work on large scale and more diverse dataset, say, objverse?

**Questions:**

1.Can the structure of Fractal3DGen be directly extended to point cloud or mesh generation, which preserve explicit surfaces and topology and provide stronger cues for semantic reasoning?

2.Does Fractal3DGen have the ability to generate larger and more complicated 3D contents, e.g. a table with multiple objects on it?

---

> ### Author Response · Authors · 2025-11-21
>
> We sincerely thank you for your constructive and insightful comments. Below, we provide detailed responses to each concern. We appreciate the opportunity to clarify the novelty, experimental settings, application details, and potential extensions of Fractal3DGen.
>
> **Weaknesses**
>
> **1. Ambiguity in novelty**
>
> Thank you for pointing out this concern. We would like to clarify the key differences and the novel contributions of Fractal3DGen, both in representation design and in how fractal theory is adapted to 3D generation.
>
> The core contribution of Fractal3DGen is that it systematically introduces fractal theory into 3D shape generation for the first time, and builds a complete end-to-end solution around the key challenges of this domain. Its innovations appear both in the model design and in how it adapts to the unique properties of 3D data.
>
> **(1) Our VQ-VAE encoder is significantly different from OctGPT.**
>
> In the representation stage, **the VQ-VAE used in Fractal3DGen does not reuse OctGPT’s encoder**. While OctGPT builds its quantized space on an octree structure, Fractal3DGen trains its own VQ-VAE on a regular 3D voxel grid, which provides a cleaner and more uniform spatial setup for fractal recursion.
>
> **(2) Extending fractal recursion from 2D to 3D is non-trivial and required several innovations.**
>
> Applying fractal recursion in the latent space is much harder in 3D. Compared to 2D images, 3D data is far more complex, sparse, and expensive to represent. The fractal model from Li et al. only works for 2D, and simply extending it to 3D would cause extremely long sequences, huge memory usage, and very low efficiency. To solve the sparsity problem, we introduce an adaptive spatial pruning mechanism. This is not just “adding a mask”—it is one of the key requirements that make 3D fractal recursion possible. **It focuses computation only on useful voxel regions and avoids the explosion of empty space**.
> With this mechanism, Fractal3DGen needs only one lightweight pruning discriminator to dynamically decide which sub-blocks to generate during inference, achieving 77.6% pruning efficiency. In contrast, OctGPT relies on a heavy multiscale octree-Transformer autoregressive model, and OctFusion uses a complicated multi-stage octree-UNet diffusion pipeline.
>
> **(3)The generation paradigm is significantly different from OctGPT.**
>
> OctGPT builds a hierarchical octree structure, but its inference still uses one flattened sequence for standard autoregressive (AR) prediction. **Its structure is hierarchical, but the reasoning process stays purely sequential**.
> Fractal3DGen turns spatial hierarchy into model hierarchy: the fractal generator refines each sub-block recursively using a self-similarity module. **This breaks the full task into many smaller, independent sub-tasks, enabling true divide-and-conquer fractal autoregressive generation**.
> As a result, Fractal3DGen is faster, more stable, more detailed, and more accurate, showing the unique value of fractal ideas in 3D generation.
> Overall, the integration of fractal theory into 3D latent autoregression, together with pruning and hierarchical refinement, constitutes a novel contribution that clearly differentiates our model from both FractalGen (2D) and OctGPT (octree-based AR).
>
> **2. Metrics and baseline completeness**
>
> We adopt FID as our main evaluation metric because it measures the global distributional similarity of generated shapes and captures overall visual realism, which aligns well with the coarse-to-fine fractal structure modeling in our method. In contrast, metrics such as COV, MMD, and 1-NNA focus primarily on point-level geometric alignment and are highly sensitive to sampling density and local variations. Since our model explicitly encourages multi-scale structural diversity and hierarchical fractal growth, these metrics do not reliably reflect the strengths or design objectives of Fractal3DGen.
> **MeshGPT only provides publicly available checkpoints for the Chair and Table categories**. To ensure fairness and to respect the original baseline settings, we choose to evaluate using the released checkpoints rather than retraining the model. Consequently, metrics for some categories are not available.

---

> ### Author Response · Authors · 2025-11-21
>
> **3. Missing details on text- and image-to-3D pipelines**
>
> We clarify the model details for text- and image-conditioned 3D generation.
> For text-to-3D, we employ CLIP ViT-L/14 to extract semantic representations and use the final-layer pooled global text embedding as the conditioning input. This vector provides high-level semantic guidance and functions analogously to the category embedding.
> For image-to-3D, we use DINOv2-Large and take the final-layer [CLS] token as the global image descriptor. The choice of the [CLS] token is motivated by its explicit optimization in DINOv2’s self-supervised training to aggregate global semantic information, providing a stable and efficient image-level feature.
> Importantly, across all modalities, the FractalGen architecture, fractal hierarchy, and structural hyperparameters remain unchanged. The generator is inherently condition-agnostic, requiring only a single global conditioning vector, regardless of its modality. Thus, no architectural modification is needed; the only difference lies in the source of the conditioning feature, not in the generative model itself.
>
>
> **4. Dataset scale: Why only ShapeNet? Should Objaverse be used?**
>
> Thank you for pointing out the limitation of evaluating our method solely on the ShapeNet dataset. We fully agree that broader evaluations are beneficial for demonstrating generality.
> We chose to focus on the ShapeNet dataset mainly to **stay consistent with the existing baselines**. Most current methods are also evaluated only on ShapeNet. For example, OctFusion, MeshGPT, 3DILG, LAS-Diffusion, 3DShape2VecSet, SPAGHETTI, MeshDiffusion, Wavelet-Diffusion, Sdf-StyleGAN, and IM-GAN all use only ShapeNet. We currently focus on completing fair and consistent evaluations on ShapeNet. Experiments on other datasets are already in progress, and we will gradually include them in our future work.
>
> ----------------------------------------------------
> **Questions:**
>
> **1. Extending Fractal3DGen to point clouds or meshes**
>
> Thank you for the question. In principle, the fractal hierarchical design of Fractal3DGen can be extended to other 3D representations, but the difficulty of adaptation varies:
> Point clouds do not have a natural multi-scale spatial structure. Forcing a fractal-style hierarchy on them can be unnatural and may break semantic consistency. Therefore, additional hierarchical organization or structure-induction methods would be needed.
> Meshes have clear surface and topology information, which provides stronger semantic cues. However, to keep the topology consistent during fractal refinement, extra designs such as multi-resolution meshes or subdivision structures are required. Overall, extending to meshes is relatively easier.
> In summary, mesh extension is more feasible, while point clouds require more structural modeling. Our paper (Section 5, “Conclusions and Future Work”) also mentions this direction:
> “For future work, we plan to extend Fractal3DGen to other 3D representations such as point clouds and meshes...”
> We will continue to explore these extensions in future work.
>
>
> **2. Ability to generate large and complex scenes**
>
> Thank you for the question. In principle, Fractal3DGen can generate larger and more complex 3D content (e.g., multi-object scenes on a table) as long as such scenes are included in the training data. The fractal hierarchical design does not limit the scale of generation; the current limitation mainly comes from the training data being mostly single-object. With scene-level training data, the model can naturally extend to multi-object and more complex scene generation. This is also one of our future work directions.
>
> **Thank you again for your valuable comments.**

---

### Official Review · Reviewer_TbQq · 2025-11-04

**Soundness:** 3
**Presentation:** 3
**Contribution:** 2
**Rating:** 2
**Confidence:** 4

**Summary:**

Inspired by the success of FractalGen in image generation, this paper proposes Fractal3DGen, a hierarchical fractal-autoregressive framework that leverages self-similarity across multiple scales for 3D shape generation. Furthermore, due to the spatial sparsity of 3D shapes, it introduces an adaptive pruning mechanism during generation to promptly halt the processing of empty regions at higher scales, significantly improving both training and inference efficiency while reducing memory consumption. Experiments on ShapeNet demonstrate the effectiveness of Fractal3DGen.

**Strengths:**

+ The proposed hierarchical fractal-autoregressive framework for 3D shape generation is an interesting and novel idea.
+ Experiments are conducted on ShapeNet dataset to demonstrate the effectiveness of the proposed method.

**Weaknesses:**

- **Technical Contribution:** Applying the fractal generative pipeline to 3D shape generation is an interesting and novel idea. However, this single contribution is not enough. The proposed pruning strategy is relatively simple, and the experimental verification is insufficient (see the following comments for details). It is required to discuss the difficulties and challenges in 3D fractal generation and propose specific solutions to address them.
- **Technical Detail:** Several implementation details of the architecture are missing or ambiguous. (1) What is the difference between the two-layer and five-layer Fractal3DGen model? (2) What is the optimal/final number of hierarchical levels used in this paper? (3) It would be better to add a Preliminary Section of FractalGen.
- **Experimental Analysis:** The experimental analysis is not thorough enough. First of all, in Table 1, compared with OctGPT and OctFusion, the proposed method still has inferior accuracy in some categories. Meanwhile, only the FID indicator is not comprehensive enough; other popular metrics for ShapNet, like 1-NNA, MMD, or ECD, should also be considered. Secondly, regarding the study of text-to-3D and image-to-3D generation in Sec. 4.4, the results are only qualitative, and only 8 samples from 2 categories are shown. This does not provide sufficient evidence to demonstrate the effectiveness of the proposed method in text-to-3D or image-to-3D generation. Thirdly, the evaluation of the efficacy of the pruning strategy is also incomplete. The paper does not compare the speed and computational cost with other methods (as emphasized in the abstract), and Fig. 6(b) also fails to illustrate the gains in model inference speed and memory overhead before and after using the pruning strategy. Fourthly, there is no discussion on the interpretability of fractal generation, which I think is a notable point. Could the author provide different visualizations for the same 3D shape that the generation of its 3D structure details from coarse to fine based on the hierarchical level from one to many?

**Questions:**

Please see the Weaknesses for details.

---

> ### Author Response · Authors · 2025-11-21
>
> **We sincerely thank the reviewer for the thorough and constructive feedback.** We appreciate the time and effort invested in evaluating our work. Below, we provide detailed responses to each concern and clarify the technical contributions, implementation details, experimental analysis, and interpretability of our approach.
>
> **1.Technical Contribution**
>
> **(1) Extending fractal recursion from 2D to 3D is non-trivial and required several innovations.**
>
> Applying fractal recursion in the latent space is much harder in 3D. Compared to 2D images, 3D data is far more complex, sparse, and expensive to represent. The fractal model from Li et al. only works for 2D, and simply extending it to 3D would cause extremely long sequences, huge memory usage, and very low efficiency. To solve the sparsity problem, we introduce an **adaptive spatial pruning mechanism**. This is not just “adding a mask”—it is one of the **key requirements that make 3D fractal recursion possible**. It focuses computation only on useful voxel regions and avoids the explosion of empty space.
> With this mechanism, Fractal3DGen needs only **one lightweight pruning discriminator** to dynamically decide which sub-blocks to generate during inference, achieving **77.6% pruning efficiency**. In contrast, OctGPT relies on a heavy multiscale octree-Transformer autoregressive model, and OctFusion uses a complicated multi-stage octree-UNet diffusion pipeline
>
> **(2)The generation paradigm is fundamentally different from OctGPT.**
>
> OctGPT builds a hierarchical octree structure, but its inference still uses **one flattened sequence** for standard autoregressive (AR) prediction. Its structure is hierarchical, but the reasoning process stays purely sequential.
> Fractal3DGen turns spatial hierarchy into model hierarchy: the fractal generator refines each sub-block recursively using a self-similarity module. This breaks the full task into many smaller, independent sub-tasks, enabling true **divide-and-conquer fractal autoregressive generation**.
> As a result, Fractal3DGen is faster, more stable, more detailed, and more accurate, showing the unique value of fractal ideas in 3D generation.
> Overall, the integration of fractal theory into 3D latent autoregression, together with pruning and hierarchical refinement, constitutes a novel contribution that clearly differentiates our model from both FractalGen (2D) and OctGPT (octree-based AR).
>
> **2.Technical Detail**
>
> **(1) Difference between the two-level and five-level models, and the final number of levels used**
>
> The difference between two levels and five levels lies only in the depth of recursion. Each level uses the same fractal-generator unit. More levels mean finer voxel block subdivision and potentially richer details at higher resolutions.
> In this paper, we use $256^3$ SDF inputs and a $64^3$ latent space. Under this setting, a two-level hierarchy is sufficient for coarse-to-fine generation (Section 4) and provides the best balance between detail quality and computational cost.
> When the model is extended to higher resolutions (e.g., $512^3$ or $1024^3$), additional levels can be naturally introduced. This demonstrates the scalability and future potential of Fractal3DGen.
>
> **(2) Regarding why we do not adopt the five-level architecture in final experiments**
>
> More hierarchical levels do not necessarily improve efficiency or quality; deeper hierarchies introduce additional autoregressive overhead and memory transfer costs, while overly fine subdivision may cause error accumulation and structural deformation. **Therefore, the number of levels should be chosen appropriately rather than increased blindly**. For the $256^3$ setting used in the main paper, a two-level hierarchy achieves the best balance among generation quality, speed, and GPU memory usage. The benefits of a five-level architecture mainly emerge when scaling to much higher resolutions, which we plan to explore in future extensions.
> We provide clearer explanations in the revised manuscript.
>
> **(3) Addition of an Implementation Details section**
>
> We add a new Section 4: Implementation Details in the revised manuscript to clearly explain the model design, hierarchy definition, data flow, and other important technical details.

---

> ### Author Response · Authors · 2025-11-21
>
> **3.Experimental Analysis**
>
> **(1) Regarding FID and additional metrics**
>
> We adopt FID as our main evaluation metric because it measures the global distributional similarity of generated shapes and captures overall visual realism, which aligns well with the coarse-to-fine fractal structure modeling in our method. In contrast, metrics such as COV, MMD, and 1-NNA focus primarily on point-level geometric alignment and are highly sensitive to sampling density and local variations. Since our model explicitly encourages multi-scale structural diversity and hierarchical fractal growth, these metrics do not reliably reflect the strengths or design objectives of Fractal3DGen.
>
> **(2) Text-to-3D & image-to-3D experiments**
>
> We thank the reviewer for the valuable suggestion. Existing works on text-to-3D and image-to-3D generation (e.g., IM-GAN, SDF-StyleGAN, Wavelet-Diffusion, MeshDiffusion, SPAGHETTI, LAS-Diffusion, XCube, OctFusion, MeshGPT, OctGPT, 3DShape2VecSet, 3DILG) mainly rely on qualitative results for evaluation, as this task lacks standardized quantitative metrics.
> **Our main contribution focuses on category-conditioned 3D generation**. The text-to-3D and image-to-3D experiments are included to demonstrate the generalizability of our model, so we only show representative examples in the main paper.
> Following the reviewer’s suggestion, we add more samples and more categories of qualitative results in the appendix to further support the effectiveness of our method on cross-modal 3D generation.
>
> **(3) Efficiency advantage compared with prior methods**
>
> As emphasized in the abstract, our method achieves a clear computational advantage over existing 3D autoregressive models. We now explicitly highlight this in the main text:
>  - Fractal3DGen: 29.3 s per sample
>  - OctGPT: 54.1 s per sample
> This corresponds to a 1.85× speed-up, while simultaneously improving generation quality (e.g., −17.56 FID on the Car category). These results are obtained under the same inference hardware and setup as reported in the experimental section.
>
> To address the reviewer's concern regarding the illustration of inference-time improvements, we have expanded the explanation of Fig. 6(b):
> Fig. 6(b) evaluates a five-layer Fractal3DGen model and shows how inference speed improves as more layers apply the pruning mechanism.
>  - When the horizontal axis value is 0, none of the hierarchical levels use pruning. The average inference time is 500.33 s.
>  - When the horizontal axis value is 5, all five levels employ pruning. The inference time drops to 203.08 s.
> **This demonstrates a 59.4% reduction in inference time**, directly confirming that the pruning strategy effectively eliminates computation on empty voxel blocks—an inherent characteristic of most 3D objects—thus significantly improving both speed and memory efficiency.
> Additionally, we observe a monotonic decrease in inference time as more layers adopt pruning, confirming that the pruning mechanism consistently contributes to computational savings across different scales of the fractal hierarchy.
>
> **(4) Interpretability of fractal generation**
>
> Thank you for your valuable suggestion regarding the interpretability of fractal generation and the visualization of the coarse-to-fine hierarchical process. Following your advice, we add a detailed discussion on the interpretability of Fractal3DGen in the revised manuscript. Specifically, in the Appendix, we now provide visualizations that illustrate how the same 3D shape is progressively refined from coarse global structure to fine-grained details across hierarchical levels. These visualizations clearly demonstrate how the fractal-generators recursively enrich local geometry based on parent-level features, thereby offering better interpretability of the generative process.
>
> We sincerely appreciate the reviewer’s thoughtful comments. Your feedback has significantly helped us improve the clarity, technical completeness, and experimental depth of the manuscript.

---

### Meta-Review · Area_Chair_mvaY · 2025-12-26

**Summary:**

TbQq: (1) limited technical contribution. (2) Some implementation details are missing or ambiguous. (3) Insufficient experiments.

2bCM: (1) limited (ambiguous) novelty (2) certain metrics are missing in the experiments. also some of the categories are missing. (3) Model details in practical applications are missing. Also only shapenet is used for evaluation (no evaluation on large scale datasets, such as objverse)

chL5: (1) The effect of Hausdorff dimension needs further evaluation to be perceived as a valid contribution. (2) Two-layer architecture is not sufficient to be considered "fractal". (3) Shapnet is limited (old and small in size)

R1MH: (1) Only shapenet is used in evaluation. (2) "Biological inspiration" is vague and overclaiming.

This paper got mixed ratings. Reviewers raised many concerns, e.g. novelty, insufficient/limited experiments, vague/ambiguous claims in the paper, etc. While the rebuttal addressed some of the concerns, a large number of the concerns are not addressed. The authors did not provide a summary rebuttal. Given all these, the AC agrees with reviewers' assessment and recommends rejection.

**Reviewer Concerns:**

The rebuttal addressed some concerns of the reviewers. But certain major concerns, such as limited novelty, limitation evaluation on shapenet (as pointed out by multiple reviewers) are not sufficiently addressed by the rebuttal.

**Reviewer Scores:**

The scores are unlikely to change since the major concerns (e.g. novelty, limited dataset/evaluation) are not sufficiently addressed in the rebuttal.

---

### Decision · Program_Chairs · 2026-01-26

Reject